# A Case Study on Curling Stone and Sweeping Effect According to Sweeping Conditions

**DOI:** 10.3390/ijerph18020833

**Published:** 2021-01-19

**Authors:** Tae-Whan Kim, Sang-Cheol Lee, Se-Kee Kil, Sang-Hyup Choi, Yong-Gwan Song

**Affiliations:** 1Korea Institute of Sport Science, Seoul 01794, Korea; burumi@sports.re.kr (T.-W.K.); sclee@sports.re.kr (S.-C.L.); kclips@kspo.or.kr (S.-K.K.); 2Center for Sport Science in Jeju, Jeju 63189, Korea; 3Department of Marine Sports, Pukyong National University, Busan 48513, Korea; 4Department of Marine Design Convergence Engineering, Pukyong National University, Busan 48513, Korea

**Keywords:** curling, stone, sweeping, broom, electromyography

## Abstract

The purpose of this study was to use the same ice temperature and air temperature as the Pyeongchang Curling Stadium by using an Ice Chamber. Then, launch the stone at the same speed, and move according to the sweeping conditions (perpendicular to the axis of motion, along the axis of motion) of male and female elite curlers. The aim is to provide sports science information required for curling athletes by analyzing the distance of the stone, the change in the speed of a moved stone, the change in broom acceleration, and athletes’ muscle activity. The results of experiments conducted on four male and four elite female curlers are as follows. Under gender, the stone’s movement distance was long after the sweeping of male athletes, and the speed of the stone was not different according to the conditions. The broom’s acceleration did not show a difference in both the sweeping condition and the athlete’s gender condition, and the muscle activity did not show a significant difference in both the sweeping condition and the gender condition. In summary, it is thought that male athletes moved the stone farther by raising the ice surface temperature by vertical load than female athletes. Also, there was no statistically significant difference in muscle activity results, but it was found that there was a specific pattern of muscle activity in the pushing and pulling motions during the sweeping of male and female athletes. It is expected to be used as primary data.

## 1. Introduction

Curling is a team sport in which two or four (mixed doubles) teams are formed to win or lose depending on which team places more stones on a specific location called House on Ice. Four players (Lead, Second, Third, Skip) will push the stone off the ice from the starting line to the house position about 20 m away (Delivery), and the stones pushed from the starting line will then move to the inertia [1].

Curling is closely related to game performance, such as the state of the ice, the initial speed and the number of rotation of the stone during delivery, and the sweeping speed and position in the sweeping movement, and scientific data and field experience play an essential role [2]. In particular, various curling factors are determined according to the environmental conditions of ice, and one of them is sweeping. Sweeping is the motion of sweeping the ice floor with a brush in the direction the stone moves. A sweeper determines the ice condition in a curling match and then intimately affects the win or loss of the match, depending on the sweeper’s ability to perform the sweeping. The surface temperature rises for a little while, reduces the stone’s rotation, and allows it to move longer [3]. To effectively perform such sweeping, it is necessary to determine the stone’s speed and direction, and requires a downward force (vertical load) and fast sweeping speed [4].

Hitoshi Yanagi et al. [5] used a self-developed brush tool to measure vertical load during the sweeping to identify the difference between the vertical load forces of the curlers and beginners and reported that the difference between the vertical load forces of the experienced and novice would apply a force of approximately 600 N.

Buckingham et al. [6] reported that the maximum force of a vertical load was about 400 N. Kim & Chae [7] reported an average vertical load force of 632.4 N on elite male athletes, and 406.6 N on female athletes was applied. Also, Marmo, Buckingham, & Blackford [8] said that as the brush head is positioned closer to the player performing the sweeping, the brush head’s vertical load increases. Therefore, it can be said that the pattern of muscle use of athletes sweeping is also essential.

Previous studies on the speed of sweeping, Marmo et al. [8] reported a speed of 2 m/s, and Bradley [3] measured the speed of sweeping for elite curlers. It appeared 4.32 times and about 3.81 times per second for women. What can be expected from this study is that when sweeping in the direction of the stone’s movement, the ice surface temperature rises, and a water film is created, which lowers the friction coefficient between the stone and the ice, increasing the distance of the stone. The trajectory can be changed [9]. In this case, it can be inferred that the maximum vertical load and sweeping speed can effectively increase the ice surface temperature. A study by Marmo et al. [10] found that strong swiping by Olympic-level male curlers increased the ice’s temperature, thus reducing the friction and that changes in this temperature are affected by the athlete’s sweeping acceleration, vertical load force, and speed of sweeping. 

As a result, adjusting the movement distance and trajectory of the stone is a decrease in friction through sweeping except for delivery, and it is considered that it is essential to know the movement patterns of athletes during sweeping. However, there is no research considering athletes’ sweeping characteristics while the ice environment is controlled in curling. Therefore, the purpose of this study is to implement the same ice temperature and air temperature as the Pyeongchang Curling Stadium using an Ice Chamber, and then to analyze biomechanically (temperature change in ice speed, length of stones moved, speed change of tons moved, change in broom speed, and influence of curling athlete’s muscle activity). As we know, it is the first kinematic analysis of curling.

## 2. Materials and Methods

This study was conducted for four male and four female elite curling athletes, and the selected subjects were only those who voluntarily participated in the study without the leader’s intervention. All curling athletes are from the Republic of Korea national team. Also, all athletes’ average training experience is 6.7 years, and the average age is 23.5 years. The selected subjects have also previously signed an agreement to use personal information and an agreement to participate in research.

The maximum sweeping frequency was controlled based on Bradley’s study [4] results (4.32 Hz for men and 3.81 Hz for women). In the sweeping method, the brush pad was moved to the maximum in the horizontal (perpendicular to the axis of motion) and vertical (along the axis of motion) directions. An IMU sensor was attached to the Broome to measure the acceleration during maximum sweeping in real-time (1500 Hz). At this time, in order to check the degree of acceleration change in real-time, data was collected in real-time using the MR3 program (Noraxon, Scottdale, Arizona, USA).

The upper left and right deltoids (LDT, RDT), biceps brachii (LBB, RBB), brachioradialis (LBR, RBR), pectoralis major (LPM, RPM), rectus abdominis (LRA, RRA), triceps brachii (LTB, RTB), extensor carpi ulnaris (LEC, REC) and latissimus dorsi (LLD, RLD) selected in this study to obtain electromyograph (EMG) data during sweeping. The skin surface hairs of a total of 16 muscles are removed and then washed with alcohol to remove bipolar surface electrodes (dual-electrode, Noraxon, USA; distance between electrodes: 1.5 cm), and the location of each muscle was selected by referring to the guidelines of Kim Tae-whan et al. (2014). Wireless EMG system (DTS Probe Transmitter, Noraxon, Scottdale, Arizona, USA, sample rates 1500 fixed, input impedance > 100 MΩ, CMRR > 100 dB, Input Range ±3.5 V, center to center distance = 15 mm) to measure actual muscle activity was used. The measured raw data is smoothed to the root mean square (RMS 50 ms) after applying the full-wave rectification to all signals after the bandpass filter 10–350 considering the research subject’s movement smoothing to analyze. Normalization is performed by cutting the start and endpoints of the pushing and pulling motions five times in the entire sweeping motion, and data is calculated based on the average value. Also, RVC (reference voluntary contraction) cuts and specifies until the athletes’ first sweeping has been pushed and pulled six times and normalized accordingly.

In order to realize the same conditions as the Gangneung Curling Center ice environment, the ice information implemented at the 2017 Test Event was collected through cooperation with the PyeongChang Organizing Committee, and the Ice Chamber used the obtained information at the Sports Goods Testing Center of the Korea Institute of Sport Science. Ice Chamber is a device to test the performance of ice equipment. It is equipment that models an ice field’s environment and drives it to measure data such as temperature, speed, friction, and rotational force. As a result of data obtained during 300 curling competitions at Gangneung Curling Center, the average air temperature was 13.3 ℃, and the average ice temperature was −4.0 ℃. Based on this, the average air temperature and the average ice temperature during the experiment in the ice chamber in this study are 13.2 ℃ and −4.2 ℃, respectively, and it can be considered that the environment is similar to the ice environment conditions of the actual stadium.

In this study, after the stone was launched (by a device that automatically shoots at a constant speed) in the ice chamber, the moving distance of the stone, the rate of change in the speed of the stone, the change in the broom acceleration, and the activity of the muscle were measured according to the sweeping method. The spin speed of the stone launcher was set to 300 rpm. Also, the calculated results were presented as mean and standard deviation, and then two-way ANOVA analysis was performed using SPSS 25.0 statistical program (IBM, Armonk, NY, USA). The level of significance at this time is *p* < 0.05.

### 2.1. Analysis of Stone Moving Distance When Sweeping

After the stone was launched, the stone’s moving distance according to the presence or absence of sweeping was measured using an along the axis of motion measuring instrument installed in the ice chamber.

### 2.2. Analysis of the Rate of Change of Stone Speed during Sweeping

After sweeping the stone, the stone’s speed during sweeping was measured using an optical speed meter.

### 2.3. Broom Acceleration Change during Sweeping

Acceleration was measured through an Inertial Measurement Unit (IMU) sensor (myoMOTION Sensor, Noraxon, Scottsdale, AZ, USA) attached to the broom when sweeping after the stone was fired. At this time, all accelerations according to the sweeping movement method were measured, and the average value of the three-dimensional acceleration variables when pushing and pulling the broom was obtained.

### 2.4. Athlete’s Muscle Activity (EMG) Changes during Sweeping

When sweeping after the stone was launched, the athlete’s electromyogram was measured according to the direction of the broom movement. For the measured EMG variable, each muscle’s average EMG value expressed at each maximum average acceleration (push + pull) was obtained. Participants provided informed consent, and the procedures were approved by the Institutional Review Board (Korea Institute of Sport Science-Seoul; KISS-1803-018-02).

## 3. Results

In this study, four male and four female curling athletes were analyzed for the average moving distance variable of the stone according to the sweeping perpendicular to the axis of motion and the sweeping along the axis of motion in the ice chamber. 

### 3.1. Stone Movement Distance When Sweeping

Two-way ANOVA was performed to verify each sweeping type and gender’s main effect for the stone movement distance and the interaction effect between the sweeping type and gender in Figure 1.

As a result of two-way ANOVA, the sweeping type’s main effect on the stone movement distance was significant. As a result of Bonferroni’s multiple comparisons, the stone movement distance along the axis of motion (M = 78.733) compared to the perpendicular to the axis of motion (M = 58.985) appeared to be high. The main effect of gender was also significant, but the stone movement distance of male athletes (M = 98.796) was higher than that of female athletes (M = 38.922).

As a result, the main effect of the sweeping type was significant (*F* = 5.085, *p* < 0.05), and the main effect of gender was also significant (*F* = 46.737, *p* < 0.01). Moreover, the interaction effect of sweeping type and gender was not significant.

### 3.2. Stone Movement Speed When Sweeping

Two-way ANOVA was performed to verify each sweeping type and gender’s main effect and the interaction effect between the sweeping type and gender for the rate of change in stone movement speed in Figure 2.

As a result, the main effect was not significant in both the sweeping type and gender concerning the change rate of the stone movement speed. Also, the interaction effect of sweeping type and gender was not significant.

### 3.3. Broom Acceleration Changes When Sweeping

Two-way ANOVA was performed to verify gender and the interaction effect between the sweeping type and gender X (Figure 3), Y (Figure 4), and Z (Figure 5), and the main effect of each sweeping type by when sweeping across X (Figure 6), Y (Figure 7), and Z (Figure 8) axis.

As a result, the interaction effect of sweeping type and gender was not significant. Also, the main effect was not significant in sweeping type and gender for broom acceleration.

### 3.4. Comparison of EMG between the Gender Condition in Pushing and Pulling Motions during Sweeping

Two-way ANOVA was performed to verify each sweeping type and gender’s main effect and the interaction effect between the sweeping type and gender for muscle activity in a sweeping push and pulling motion (Figure 9 and Figure 10). 

As a result, there was no significant effect on muscle activity in both sweeping type and gender. Also, the interaction effect of sweeping type and gender was not significant.

## 4. Discussion

The present study is the first to examine the kinematics of curling stones and sweeping skills. In preparation for the 2018 PyeongChang Winter Olympics, this study implemented an Ice Chamber for four male and four elite female athletes to measure the distance and speed of movement of the stone according to the conditions of sweeping, changes in acceleration of the broom during sweeping, and the muscle activity of athletes.

### 4.1. Stone Movement Distance When Sweeping

In male athletes’ sweeping, the distance of the stone’s movement appeared to be longer than that of the stone’s in women’s sweeping, which is believed to have a higher sweeping effect than that of the male athletes since there was no difference in the sweeping. In the previous study, Kim et al. (2013) reported that sweeping along the axis of motion makes the ice surface temperature higher than sweeping the perpendicular to the axis of motion, as a result of verifying the effect of sweeping type through the automatic sweeping device is consistent with the study results that appear longer in sweeping in the along the axis of motion direction than in the perpendicular to the axis of motion even in this study.

### 4.2. Stone Movement Speed When Sweeping

The rate of change of stone speed after sweeping of male athletes was slightly lower than that of female, and it was found that the sweeping effect had some influence on the rate of change of speed of the stone. However, since there was no statistically significant difference, it is considered that the reliability of the difference between the genders of athletes for the sweeping effect is not high.

### 4.3. Broom Acceleration Changes When Sweeping

Statistical significance of broom acceleration during sweeping was not found in all three conditions: gender, sweeping type, and gender x sweeping type. In terms of gender, female athletes showed a higher acceleration in both X, Y, and Z directions of pushing and pulling. In the sweeping type, the sweeping perpendicular to the axis of motion in both the X, Y, and Z directions of the pushing and pulling motions showed higher acceleration than the sweeping along the axis of motion. Summing up the results above, broom acceleration was higher when sweeping perpendicular to the axis of motion with female athletes, but the movement distance and frequency of the stone showed better results along the axis of motion sweeping with male athletes. Therefore, it is judged that the effective use of muscle for the vertical load of broom, not the acceleration of broom, influenced the stone’s movement distance. A study by Lee & Song [11] stated that the more considerable the contact pressure of sweeping brush in front of the direction in which the stone is progressing, the more the water film is formed on the ice surface and water film lowers the friction between the stone and the ice. In the study, it was said that the higher the vertical load of the sweeping. According to Buckingham et al. [6], consistent sweeping of curling sweeping is attributed to high vertical load forces.

### 4.4. Changes in Muscle Activity of the Athletes during Sweeping

#### 4.4.1. Comparison of EMG in the Pushing Motion during Sweeping

When sweeping, both male and female athletes showed almost similar muscle activity patterns in the push motion, and there was no statistically significant difference. However, in the right extensor carpi ulnaris, male athletes showed about 21% RVC higher than female athletes. In curling sweeping, the motion should be set vertically as much as possible, and then pulled and pushed with the left arm (lower arm) should be performed more flexibly. The extensor carpi ulnaris extends the wrist with the ulnar deviation and thus has a tight grip of the wrist along with other extensor muscles around the wrist [12]. Therefore, it is judged that male athletes are making more activity in the extensor carpi ulnaris to create a large vertical force downward by strongly holding the upper part of the brush with the right hand.

Meanwhile, the female athlete showed about 11% RVC higher activity than the male athletes in the left deltoid, while the left deltoid performed a sweeping motion and moved the body and brush toward the direction of the stone and pressed the brush and simultaneously. It is presumed that this result would be due to the need to conduct abduction in the direction of progress. It is based on the principle that the deltoid and infraspinatus muscles abduct the upper arm [13]. As a result, since female athletes have relatively weak muscle strength compared to male athletes, it is judged that a compensatory muscle usage pattern appeared to use sweeping faster.

Although there was no statistical difference between the perpendicular to the axis of motion and the along the axis of motion in the swiping motion during sweeping, muscle activity of the left deltoid and right pectoralis major was higher in the along the axis of motion than perpendicular to the axis of motion (about 19% RVC and 18% RVC, respectively). Sweeping along the axis of motion has a vertical shape for the moving direction of the stone compared to sweeping perpendicular to the axis of motion, so it can be considered that the contraction of the left deltoid is essential for sweeping a larger area. It is because it is medially rotated with the clavicular portion [12]

#### 4.4.2. Comparison of EMG in the Pulling Motion during Sweeping

There was no statistically significant difference in the pulling motion during sweeping. However, the female athletes in the muscle activity between genders were higher in the right deltoid, right brachioradialis, and left latissimus dorsi than in the male athletes (in order, about 28% RVC, 20% RVC, 22% RVC). The deltoid on the right is medially rotated for the direction of the stone. Afterward, the brachioradialis on the right side contracted and flexed its arms when performing a sweeping brush pulling motion for faster acceleration than male athletes. Then, the contraction of latissimus dorsi performed a faster sweeping pulling motion. The contraction of Latissimus dorsi is to perform scapula retraction while maintaining medial rotation of the left arm holding the brush underneath. Therefore, it is a muscle activity pattern that appears to compensate for female athletes’ relatively weak vertical load.

There was no statistical difference between perpendicular to the axis of motion and along the axis of motion in the pulling motion during sweeping, but muscle activity was higher in sweeping along the axis of motion in all muscles except the right biceps brachii and pectoralis major. It results from sweeping along the axis of motion rather than sweeping perpendicular to the axis of motion to make a more vertical load. In particular, muscles with higher muscle activity during sweeping along the axis of motion are (1) the right deltoid, (2) the left rectus abdominis, (3) the right triceps brachii, and (4) the left extensor carpi ulnaris (in order, about 23% RVC, 21% RVC, 23% RVC, 27% RVC).

Overall, there was no statistically significant difference in the interaction between gender, sweeping type, and gender x sweeping during the sweeping pushing and pulling. However, the vertical load is considered the decisive factor affecting the moving distance of the sweeping stone. Naturally, the difference in the speed of the stone, the acceleration of the broom, and the muscles’ activity appearing during sweeping do not show any difference performed by all athletes. In particular, in the acceleration of the broom, female athletes appeared to show higher muscle activity patterns than the male athletes. However, the stone’s short movement distance can be said to be the result of the higher vertical load of the male athlete raising the ice surface temperature. Also, although there is little difference, it is thought that the rate of change of the movement speed of the stone is lower than that of male athletes, which may support the previous results.

## 5. Conclusions

In curling sweeping, there was no significant difference in the rate of change of the stone’s movement speed, the sweeping acceleration, and the muscle activity of the athletes, excluding the movement distance of the stone. Although not measured, there was no specific difference between the type and gender of sweeping, so the importance of vertical load force in sweeping related to the ice surface temperature can be considered an implication of this study. It suggests the importance of vertical load force when performing effective sweeping in front of the stone’s movement direction. It is also encouraging that the pattern of muscle activity when pushing and pulling during sweeping was examined. Although this study used a small number of subjects, it is judged that the data’s reliability is high because they are athletes of the national curling team. Nevertheless, analyzing the results by analyzing higher quality data through the examination of more athletes is a necessary implication for this paper. Based on this study, it is expected to be used as the primary data for muscle function training and sweeping-related research of curling athletes. In the future, if the vertical load force and ice surface temperature were additionally measured, a more reliable causal relationship could be found. Finally, since it is the first study to measure muscle activity during curling sweeping, it is considered that subsequent studies are critical.

## Figures and Tables

**Figure 1 ijerph-18-00833-f001:**
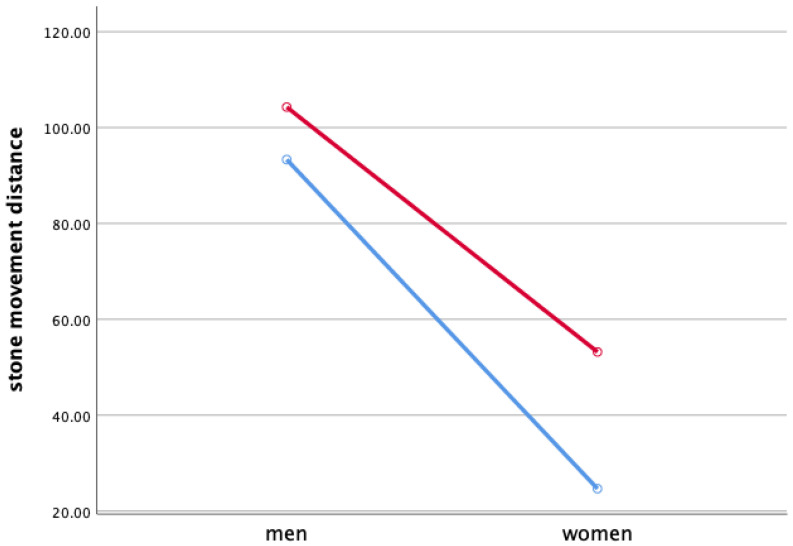
Stone Movement Distance When Sweeping (cm). Red line: along the axis of motion, blue line: perpendicular to the axis of motion.

**Figure 2 ijerph-18-00833-f002:**
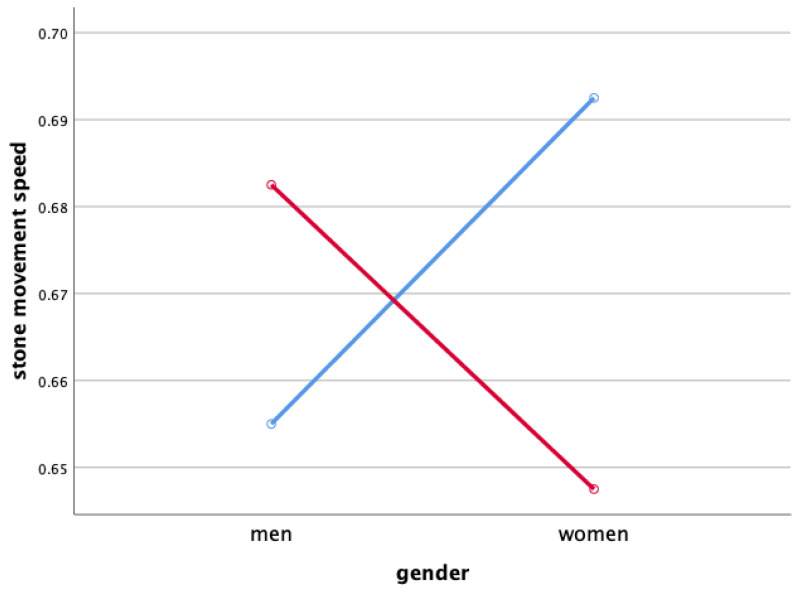
Stone Movement Speed When Sweeping (m/s). Red line: along the axis of motion, blue line: perpendicular to the axis of motion.

**Figure 3 ijerph-18-00833-f003:**
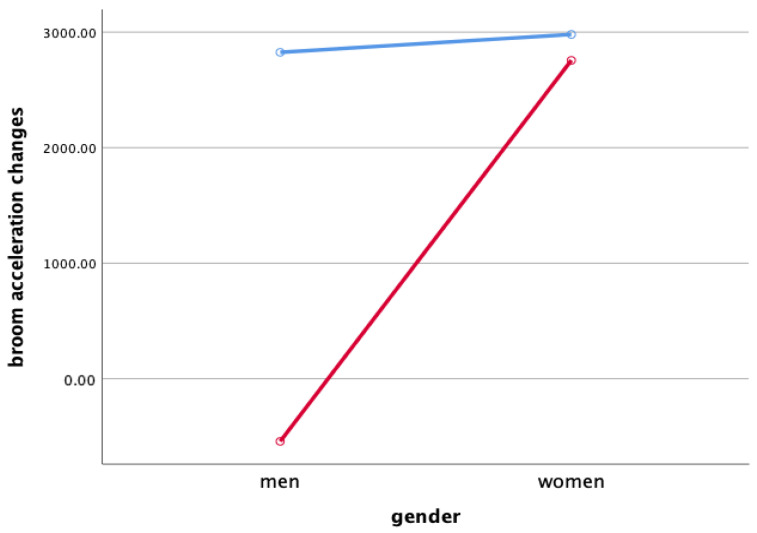
Broom Acceleration Changes When Sweeping(X)—interaction effect (mG). Red line: along the axis of motion, blue line: perpendicular to the axis of motion.

**Figure 4 ijerph-18-00833-f004:**
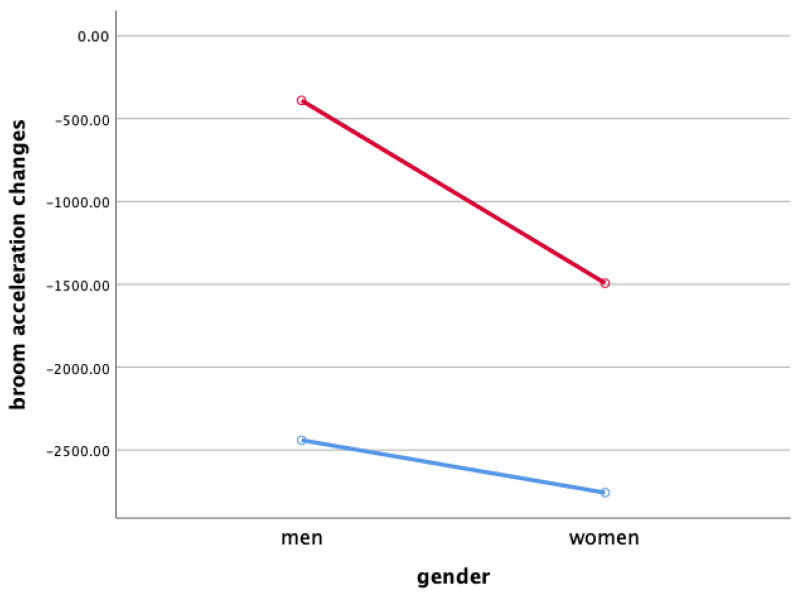
Broom Acceleration Changes When Sweeping(Y)—interaction effect (mG). Red line: along the axis of motion, blue line: perpendicular to the axis of motion.

**Figure 5 ijerph-18-00833-f005:**
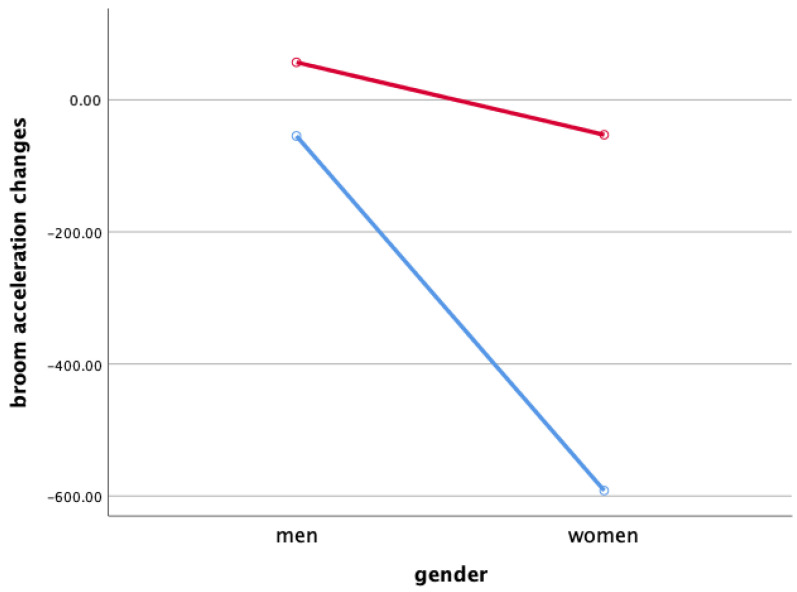
Broom Acceleration Changes When Sweeping(Z)—main effect (mG). Red line: along the axis of motion, blue line: perpendicular to the axis of motion.

**Figure 6 ijerph-18-00833-f006:**
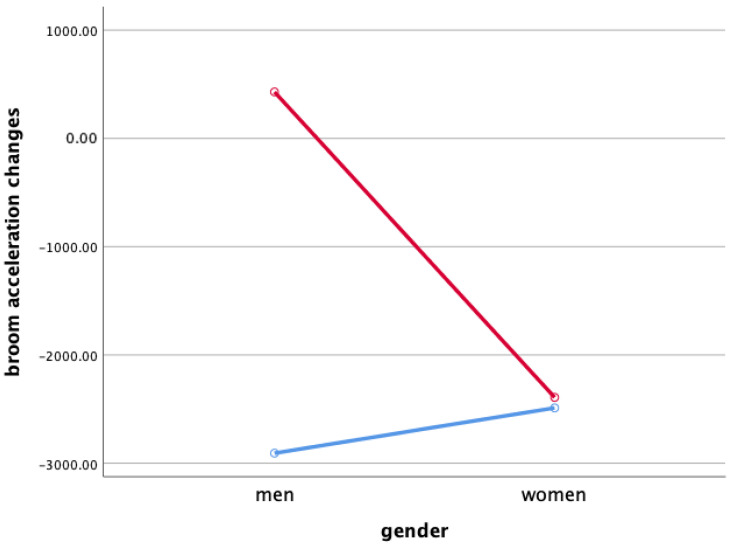
Broom Acceleration Changes When Sweeping(X)—main effect (mG). Red line: along the axis of motion, blue line: perpendicular to the axis of motion.

**Figure 7 ijerph-18-00833-f007:**
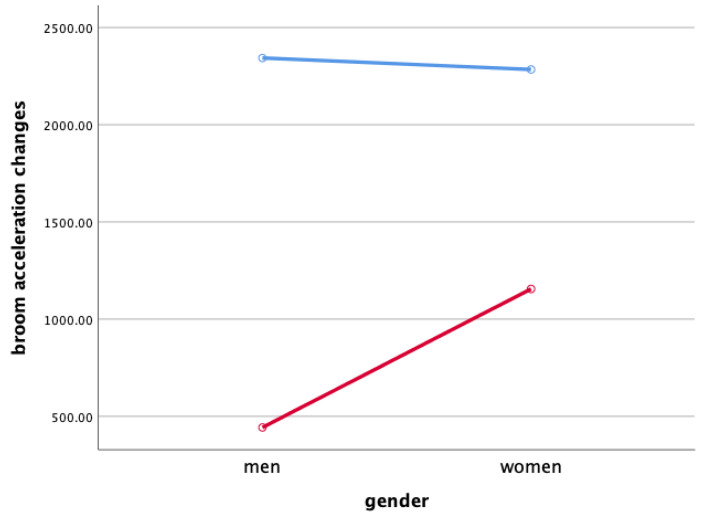
Broom Acceleration Changes When Sweeping(Y)—main effect (mG). Red line: along the axis of motion, blue line: perpendicular to the axis of motion.

**Figure 8 ijerph-18-00833-f008:**
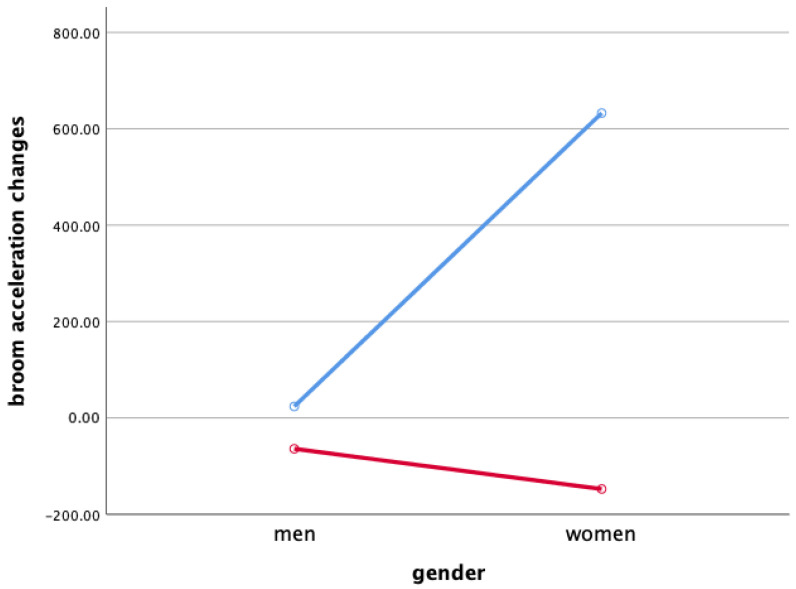
Broom Acceleration Changes When Sweeping(Z)—main effect (mG). Red line: along the axis of motion, blue line: perpendicular to the axis of motion.

**Figure 9 ijerph-18-00833-f009:**
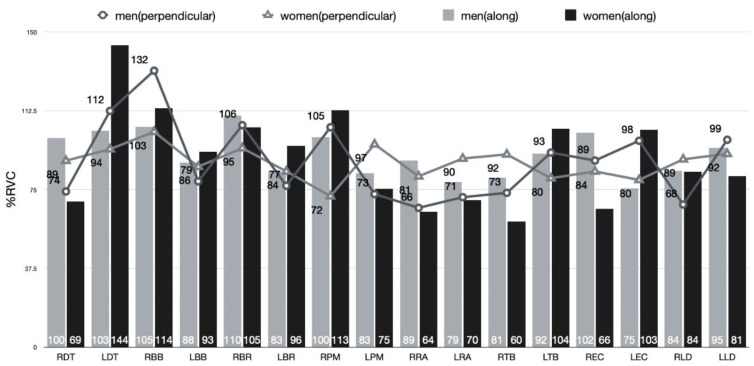
EMG in the pushing motion during sweeping (perpendicular to the axis of motion and along the axis of motion).

**Figure 10 ijerph-18-00833-f010:**
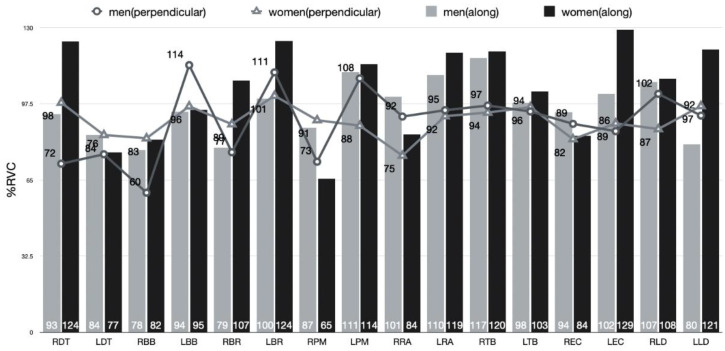
EMG in the pulling motion during sweeping (perpendicular to the axis of motion and along the axis of motion).

## Data Availability

The data that support the findings of this study are available from the corresponding author upon reasonable request due to ethical and privacy restrictions.

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
