# Peer review of "A Case Study on Curling Stone and Sweeping Effect According to Sweeping Conditions"

_ijerph, 2021, doi:10.3390/ijerph18020833_

Round 1

Reviewer 1 Report

The popularity of curling in the world is increasing and the number of studies trying to describe the parameters of both the environment and, to a very limited extent, the parameters of the competitor is increasing. The latter tries to describe the study under consideration. This has the character of a case study and therefore it is necessary to establish a working hypothesis. Given the number of people, it is necessary to adjust the conclusions, which must have the character of indicating the next direction of research activities. The study needs to be supplemented with work limits. The discussion is highly descriptive and does not discuss the causes of findings. Minimal differences between men and women are certainly given by the environment, but in addition to EMG parameters, it will be necessary to discuss the influence of muscle morphology on the monitored variables. It will also be necessary to supplement the reliability of the monitored variables.

Author Response

Thank you for your recent editorial decision letter and reviewer on our manuscript.

"I found your work to be improved upon revision and to adequately address review concerns."

As you said, the conclusion was described in consideration of the limit of the number of subjects. Also, figures 9 and 10 have been inserted for the reliability of the EMG discussion.

Although the number of subjects is insufficient, this study is based on the results of the national team curling athletes' experiments, so we think that this study has some meaning.

Please, check the revised contents in the attached file.

Thank you.

Reviewer 2 Report

Dear authors,

Thank you for the opportunity to review your manuscript. This reviewer agrees with your statement that sweeping plays an essential role within the game of curling. Furthermore, this reviewer is cognisant of the difficulty to write manuscripts in English when it is a second language. As such, please accept this review as a means to improve your manuscript for future publication.

General comments

  1. There are numerous ambiguity and grammatical errors within the manuscript that makes it extremely difficult to understand/comprehend the message that the authors are trying to convey. Thorough editing of the manuscript is needed in order to be comprehensible.
  2. The method section should have more details. How were the rocks thrown? What was the variability in the onset velocity of the rocks? What was the pebble status of the ice (was it fresh pebble or was the ice prepared in other ways than temperature and humidity)? What IMU was used on the broom? It is stated that 16 muscles were investigated. Please place them into the same sentence (the 8 different muscles are spread over two sentences). What was the EMG normalization method? How were the outcome measures calculated or normalized?
  3. It is suggested to use sweeping along the axis of motion (length) and perpendicular to the axis of motion (width).
  4. What was the global coordinate system orientation since the results are reported along the X, Y, and Z-axis? As such how important are the vertical accelerations of the broom?
  5. Frequency and speed are two different components of sweeping make sure that you use the proper terminology.
  6. What was the number of participants for the study? It is either 8 (Line 79) or 10 (Line 140).
  7. In the results section, all the graph should have appropriate labels that also incorporates the measurement units.
  8. When doing ANOVA with two factors, the interaction effect should be the first one reported.
  9. Consider merging the figures into figures with multiple panels to consolidate the concepts.
  10. The presentation of some raw or less processed data would enhance the comprehension of the manuscript.
  11. In the discussion, there is a rather large component related to the EMG results without the appropriate presentation of data within the results section. EMG results are reported in the discussion section, but it is difficult to evaluate the effect.
  12. What is the meaning of RVC?

Best regards,

Author Response

Thank you for your recent editorial decision letter and reviewer on our manuscript.

"I found your work to be improved upon revision and to adequately address review concerns."

1. We corrected the grammar overall.

2. The launch method, launch rpm, temperature, and humidity of the ice floor were presented, and the name of the sensor and the modification of the EMG regularization were made. Also, We modified the sentence about muscles.

3. We revised all the words told us.

4. With the subject looking at the stone, x represents left and right, y represents front and rear, and z represents vertical. Furthermore, the broom's vertical acceleration is not a very important factor in terms of acceleration, as it is a form of harder pressing on the ice surface when sweeping.

5, 6, 7, 8. We modified it.

9. It was difficult to merge the interaction and main effects. Instead, we added graphs for the EMG result as a merge method.

10, 11. Added EMG-related data values in graph format.

12. RVC stands for reference voluntary contraction and has been added to the method.

Please, check the revised contents in the attached file.

Thank you.

Reviewer 3 Report

The work sent for review is innovative. The scientific problem is described to a small extent in the literature. The work is generally methodologically correct. However, it has some shortcomings that must be eliminated.

Title

The title should be supplemented with the information that it is a case study. There were no criteria for selecting the muscles whose EMG activity was recorded.

Methodological part.

Description of the group of respondents: The work is devoid of detailed information about the respondents. Lack of information on the sports level, training experience, or demographic and anthropometric data of the respondents. Only 4 subjects of different sex (2 men and 2 women) do not provide grounds for generalizing the research results.

The authors do not report on the statistical methods they used to analyze the research results. They don't provide statistical software.

Results

The research results were described correctly, taking into account that the work is a case study, the detail of the description is justified.

Discussion

Merging into one part of Discussion and Implication is not a good solution. Separated from the rest of the text, the implications are difficult. I recommend separating the discussion part and transferring the implications to Practical Recommendations.

Conclussion

Conclussion is very modest in terms of lyrics. The information contained therein is too general. The statement "Based on this study, it is expected  that it can be used as the primary data for muscle function training and sweeping-related  research of curling athletes" requires more extensive justification in this part of the paper I recommend supplementing the text of the work, refining the methodological part and re-editing the discussion with the separation of practical recommendations. The title should inform you that we have a case study at work.

The text needs to be corrected as recommended. In my opinion, both the topic and the content of the paper are not in line with the topic of Exercise Medicine in Health and Disease https://www.mdpi.com/journal/ijerph/special_issues/exercise_medicine_health_disease because there are no references to the subject of this edition.

Author Response

Thank you for your recent editorial decision letter and reviewer on our manuscript.

"I found your work to be improved upon revision and to adequately address review concerns."

We have revised the general information have suggested.

I hope you understand that public health is one of primary area that include physical activity and sports.

Thank you.

Round 2

Reviewer 3 Report

The authors have included some of the suggested additions to the text There are no expected changes in:

Methodological part. Description of the group of respondents: The work is devoid of detailed information about the respondents.

Lack of information on the sports level, training experience, or demographic and anthropometric data of the respondents. Only 4 subjects of different sex (2 men and 2 women) do not provide grounds for generalizing the research results. The authors do not report on the statistical methods they used to analyze the research results. They don't provide statistical software.

Discussion Merging into one part of Discussion and Implication is not a good solution.

Separated from the rest of the text, the implications are difficult. I recommend separating the discussion part and transferring the implications to Practical Recommendations.

No justification for omitting the above-mentioned suggestions of the reviewer

Author Response

Thank you for your recent editorial decision letter and reviewer on our manuscript.

We sincerely apologize for not being able to answer all of your valuable comments and omissions. First of all, we wrote the players' average career and age (page 3, lines 84-85). Statistical techniques and programs were also entered (page 3, lines 127-130). Unfortunately, we did not measure anthropometric data such as the athletes' body length. We ask for your generous understanding of the judges.

We also moved the implications described in the discussion section to the conclusion (page 12, lines 330-331, 333-336, 338-341).

Once again, we apologize for the omission of your review and thank you for your valuable comments.